# A Pipeline-Based Approach for Object Detection on Resource Constrained Internet of Things Devices

## Abstract

Object detection with computer vision and convolutional neural networks on resources constrained devices can be challenging. The limited power and processing capacity of these devices complicates the use of deep neural networks and other object detection methods. To address this problem, we propose a pipeline-based approach. We introduce a multi-step detection pipeline considering the size of the objects to be detected and the correlation among them. To evaluate the performance of this approach, we test it in a collaborative smart surveillance system employing edge computing and the internet of things paradigm. Additionally, field testing was conducted considering real world surveillance scenarios. Results showed that the introduction of the pipeline-based processing improved the execution time by a factor of 3 and produced a significant improvement on the mean average precision.

## 1 Introduction

Ensuring safety is a major concern in metropolitan areas across the world. With the growth of smart cities, an innovative solution for combatting crime is the implementation of smart video-surveillance systems. They have the ability to automatically detect potential threats and quickly notify law enforcement agencies, providing a more effective approach to crime prevention.

Smart surveillance systems commonly employ the internet of things (IoT) paradigm and are embedded both on edge and cloud devices. Therefore, the major challenges faced by such systems are: resources management at the edge, bandwidth management and latency of critical messages. In the present work we propose a pipeline-based approach for object detection on resources constrained IoT devices and apply the methods in a surveillance system. In our work, we define "pipeline" as the sequence of all linked steps taken by the systems for the capture and processing of all necessary information for its operation.

A pipeline was carefully designed aiming processing efficiency and classification accuracy, implemented and evaluated with real world testing in partnership with the São Paulo Police Department.

Leveraging our approach with a surveillance system results in advantages like mobility, amount of detected objects/threats and automatic notifications. Moreover, the main contributions of our work are listed below:

- A pipeline-based approach to enable multi-class object detection on resource constrained edge devices.

- AI Pipeline: a sequence of inference steps optimized to satisfy the trade-off between processing time and precision.

- Real world tests (TRL 6): demonstration of the system working in the field, installed in police vehicles.

## 2 RELATED WORK

The existing IoT smart video-surveillance systems are capable of object and event identification by employing state of the art methods in computer vision (CV) and artificial intelligence (AI). Many of them are proposed for edge computing, cloud computing or, more commonly, for a hybrid edge-cloud collaboration. Recurrent challenges of these types of systems are: resources management at the edge, bandwidth management and latency of critical messages. One strategy to tackle these problems is to carefully design the pipeline steps. In this section we analyse the pipeline flow of recent systems, specifically the steps taken to increase the efficiency, detection accuracy and bandwidth saving.

Panganiban et al. (2022) propose an IoT license plate recognition system based on three different pipeline approaches: edge-heavy, cloud-heavy and hybrid. The general flow for all three pipeline is to firstly detect, in the video feed images, regions of possible license plate and forward them to the next step, where the recognition of characters is performed. In the edge-heavy pipeline both steps are performed on the edge devices and the results are stored in the cloud. On the contrary, in the cloud-heavy pipeline all steps are performed in the cloud. This approach requires a larger bandwidth (BW), since all video frames are streamed to the cloud. The hybrid pipeline strategy is to perform plate detection in the edge and sent a cropped image containing the region of interest to the cloud, where plate recognition is performed. The evaluation metric adopted to evaluate all pipeline approaches was the capture-to-result time in seconds (CTR). For a low BW (¡1600 kbps) and less than 4 edges nodes, the hybrid pipeline performed better with a CTR of 15 s. However, for more than 4 edge instances, the edge-heavy pipeline performed better with a CTR of 10 s. The cloud-heavy pipeline achieved the best CTR of 5 s, but required a BW of 2,500 kbps and less than 4 nodes.

Authors Ke et al. (2021) describe an IoT parking occupancy estimation system based on CV and AI. The strategy of the proposed system is to split the computational load between edge devices and servers, targeting optimal system performance. The first step of the proposed pipeline is to manually label parking spaces on the server side and then apply a matching algorithm according to the vehicle positions, which are detected on the edge side. The pipeline also considers normal or low visibility conditions, e.g, foggy or rainy weather. In the second case, the pipeline approach is to combine two detection methods: a Mobilenet CNN model with single shot multibox detector (SSD) and a Background modeling detector (BG). Instead of images, the forwarded results are the detected bounding boxes and position, reducing data volume in the network. The system evaluation was done in real world scenarios at parking garage. Considering several weather conditions, the achieved average accuracy was 95,6 %.

The work proposed by Fathy & Saleh (2022) describes an IoT smart video-surveillance system, capable weapon detection and automatic notifications of events. Object detection and classification (e.g. firearm, knife, phone, card) is performed by edge devices with light YoLov5 models (v5n, v5-lite e and v5-lite s ). Among the system's pipeline steps, it is proposed a Software-Defined Networking (SDN) for more efficient network usage, controlling bandwidth and speeding up critical notifications. Therefore, in this case, the pipeline approach is to control the network rather than modifying the detection steps. The evaluation of the proposed adaptive QoS model revealed improvements in performance in terms of jitter, packet loss, and average throughput. The evaluation of the light CNN models employed at the edge devices revealed the YoLoV5n to perform the best, achieving a mAP of 95 % for pistol detection. However, the system was not evaluated in real world scenarios.

Sultana & Wahid (2019) propose a smart surveillance system for home usage. The architecture of the system include edge nodes installed throughout a house and fog-nodes, which can control several edge instances. The first step of the pipeline flow is to detect motion on the edge side with pixel-based background subtraction techniques, which can be performed nearly instantly. In case motion is detected, the video stream is forwarded to a fog node, where firearm and knife detection are performed with VGGNet. Lastly, upon a detection of a crime, automatic notifications are sent by the fog servers to the authorities. By employing this pipeline design the systems saves energy, BW and CPU. Object detection is performed in 15 s at the fog and the system total operation time is 18 s.

The system entitled Hawk-Eye (Ahmed & Echi, 2021) can detect multiple classes of threats, such as weapons, vehicles and masked people. Two different pipeline flows are proposed: the first is evaluated in edge devices and the second in the cloud. In both cases, the pipeline initial flow is to detect motion with a background subtraction method. Next, objects are detected and classified with a neural network. At the cloud, a Mask R-CNN model was built, enabling the system to make a high-quality segmentation mask for each object in the images. A lighter CNN was employed at the edge, enabling the system to detect and classify objects locally, without relying network availability. Regarding object classification, no further steps are taken in the pipeline. The achieved results were different for the cloud and edge-based pipelines: the prediction time for pistols was 4.1 s with the R-CNN (cloud) and 5 ms for CNN (edge).

As can be seen above, among the recent investigations in smart surveillance, many are interested in weapon detection, since firearms and knives can indicate a severe security threat. Some of these works propose interesting pipeline approaches to increase the recall and reduce the false positive rate of weapon detection. Ruiz-Santaquiteria et al. (2021) propose an AI-based method for weapon detection, which combines both object appearance and human body pose. Similarly, Castillo Lamas et al. (2022) describe a weapon detection system based on human pose estimation, which aims to mitigates false positives that can arise in systems based on weapon appearance exclusively. In both systems the additional step in the pipeline flow led to an increase in the weapon detection mAP.

Cob-Parro et al. (2021) proposed a IoT smart video surveillance system specifically for edge computing. The AI-application uses a MobileNet-SSD architecture and is capable of detecting, tracking and counting people. Due to the limited processing power of the edge nodes, the researches were interested in the relation between performance and energy consumption of the system. The pipeline-based strategy to increase the performance was to use parallel processing of multiple video streams, which was possible due to the VPUs present in the edge devices. The inference computational cost for the algorithm using a CPU and a VPU was, respectively, 13.93 ms and 8.71 ms. Regarding people detection, no specific steps are taken in the pipeline, other than the standard MobileNet-SSD methods.

Chen et al. (2022) describe a video surveillance system for smart cities. The authors propose an IoT edge-cloud collaboration system, capable of classifying multiples classes of large objects, e.g. vehicles and bicycles. The first step in the pipeline flow is to perform, at the edge, object classification with YoLo and foreground estimation (image matting). Secondly, the extracted foreground objects are then compared with those classified by the CNN. Next, the objects that can not be automatically classified are sent to a cloud AI system, where they are manually labeled and used to retrain the CNN. Lastly, the final step in the pipeline is to update the model at the edge devices, increasing the object classification capability of the system. Across all classes, the achieved mAP was 0.983 with YOLOv4.

In addition, the industry has also shown interest in mobile smart surveillance system. For instance, Neto et al. (2018) proposed a fog-computing based system capable of crime detection in public bus services. The system can classify events in real-time and generate automatic notifications upon the detection of predetermined threats, warning the competent authorities. The first step considered in the pipeline is to pre-process the images with edge devices in-vehicles. Next, object classification is done in the cloud, which is also responsible to notify authorities upon detection of crimes. Similarly, De Biase et al. (2020) propose a collaborative and mobile surveillance system embedded in normal vehicles. The system uses edge computing for threat identification and automatic warning notifications.

We noticed from the aforementioned works that systems often consider strategic steps in the pipeline design to tackle recurring challenges of IoT-based smart surveillance. Most systems split the computational load between edge and cloud and some systems reduce data volume, optimizing the network usage. However, none is focused on overcoming such challenges by a pipeline-approach uniquely, specially when considering multi-class object detection on edge devices without GPUs and without cloud assistance. Therefore, we propose a pipeline-based approach for object detection on resource constrained IoT devices and evaluate it in a surveillance system embedded in single-board computers, which will be described in the following sections.

## 3 PROPOSED ARCHITECTURE

### 3.1 TRAINING PROCESS

In this section we describe the training process phase, where convolutional neural networks (CNNs) were trained to classify objects from regular RGB images. We trained the CNNs YOLOv3 (Redmon & Farhadi, 2018) and YOLOv4 (Bochkovskiy et al., 2020) to classify the following objects:

1. People.
2. Firearms.
3. Vehicles.
4. License plates.
5. License plate's characters.

We used our own built dataset, composed by 135,000 labelled images, which includes real images of vehicles (cars, motorcycles and bicycles), real and synthetic images of weapons, real and synthetic images of license plates. Moreover, real images of firearms were labeled in a manner to reduce the interference of human body parts.

Since the events classifications should be done with edge computing, we evaluated the following light models:

1. YOLOv3 Tiny (2 detectors YOLO)
2. YOLOv3 Tiny 3L (2 detectors YOLO)
3. YOLOv4 Tiny (2 detectors YOLO)
4. YOLOv4 Tiny 3L (3 detectores YOLO)

The metrics employed for the models performance evaluation are based on the classic confusion matrix. More specifically, we employed the precision for each class ($P$), the accuracy or total number of correct prediction ($A$) and the recall ($R$). These metrics are defined as follow:

1. $P = \frac{TP}{TP+FP}$
2. $A = \frac{TP+TN}{TP+FP+TN+FN}$
3. $R = \frac{TP}{TP+FN}$

Where,

- TP: True Positive
- FP: False Positive
- TN: True Negative
- FN: False Negative

The goal for firearm detection was to achieve a mean average precision of (mAP) of at least 80 %. Initially, a first round of training was performed with standard parameters. Based of these first results, more refined training rounds were conducted. Data augmentation methods like mosaic, mixup and blur were applied and evaluated. Finally, for designing the detection pipeline, different input images dimensions were evaluated according to object distance from the camera.

### 3.2 EMBEDDED INFERENCE PIPELINE

The mechanism for embedded event detection is based on a pipeline of object recognition steps, which optimizes both processing time and accuracy. There are two main principles behind the pipeline-based approach:

1. Large objects are easier to detect. This makes it possible to re-scale the input images to significantly low resolutions, which in turn reduces inference time. Specifically, reducing resolution to half of the original cuts the overall amount of pixels to $1/4$, resulting in a much smaller inference time.

2. The presence of some objects are correlated. For example, since license plates are always attached to a vehicle, there is no need to try to detect license plates in the absence of a vehicle. This enables rapidly dropping uninteresting frames, thus improving the rate of frame processing.

A step of the pipeline can be defined by a function of the following form:

$P_{step}(f, p) \rightarrow r$

Where:

- $f$ is the captured frame from the camera, in its original resolution. Using the original frame in every step enables extraction of the bounding box for detected objects while preventing build-up from consecutive resizing operations.

- $p$ contains the result of the previous step, such as detected items and respective bounding boxes. This item will be $nil$ (empty) in the first step of the pipeline.

- $r$ contains the result of a particular step. It can be logged, transmitted, or passed to a subsequent step of the pipeline.

Finally, the connection between the steps is done via an external orchestrator module, which is application-specific.

### 3.3 USE CASE: DETECTION PIPELINE FOR A SURVEILLANCE SYSTEM

Considering a surveillance system, a four-step pipeline was assembled for the detection of three types of events: crowds, guns, and license plates. The detection pipeline is defined below, as well as shown visually in Figure 1.

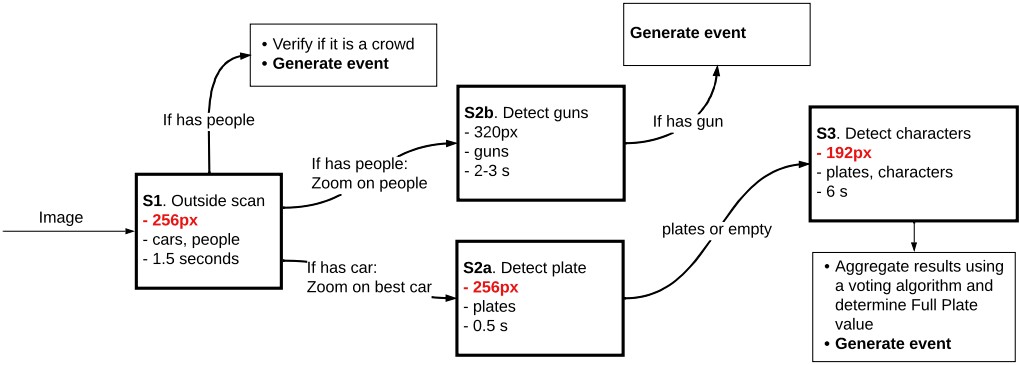

Figure 1: The proposed detection pipeline for a mobile surveillance system.

$P_{scan}(f, nil) \rightarrow r_{scan}$

$P_{gun}(f, r_{scan}) \rightarrow r_{gun}$

$P_{plate}(f, r_{scan}) \rightarrow r_{plate}$

$P_{chars}(f, r_{plate}) \rightarrow r_{chars}$

In the first step, $P_{scan}$, the original frame is down-scaled and fed into a network which searches only for two categories: person and car. If at least one person is detected, the system will verify whether there is a crowd, and in that case, generate a crowd event; it will also zoom into the largest person

box (closer to the camera), and feed it to the $P_{gun}$ step. If the latter finds a gun within the cropped person image, it will generate a gun event.

Returning the the $P_{scan}$ step, if it finds a vehicle, it will zoom into the vehicle box and feed it to the $P_{plate}$ step, which will look for license plates. If a plate is found, it will again be zoomed into, and passed to the final $P_{chars}$ step. In this step, the characters are extracted and submitted to a heuristic which verifies whether they constitute a license plate, in which case a license plate event is generated.

## 4 IMPLEMENTATION

The proposed system was implemented in real hardware and tested in the field, reaching a technology readiness level of prototype demonstration in relevant environment (TRL 6).

### 4.1 HARDWARE

The main component of the hardware solution is the Labrador 64[1], a 1,3GHz quad-core arm-based single-board computer with 2 GB of volatile memory and 16 GB of flash. Connected to the Labrador, are:

- USB Action Cam: camera used to capture high-resolution frames in a mobile environment.
- Pulga Stack: a modular subsystem composed of a cortex-M4F -based microprocessor, a GPS module for time and location, and a LoRaWAN module for event transmission. The latter is also connected to a dipole 915 MHz antenna. The Labrador and the Pulga Stack are connected via a four-wire flat cable, as shown in 2a.
- Power cable: a cable for powering the system with a 12 V power supply. On the Labrador end, it features a P4 connector; on the other end, it features a automobile auxiliary power plug [2].

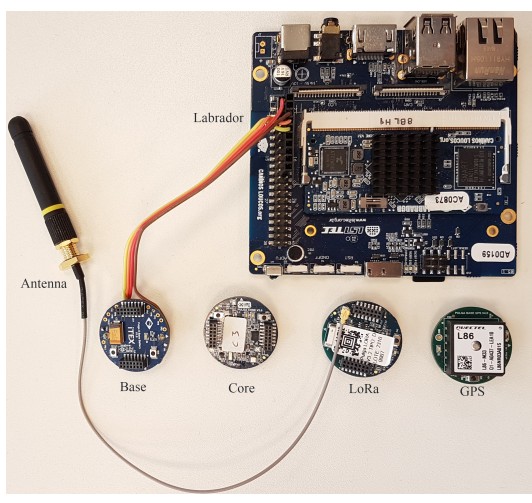
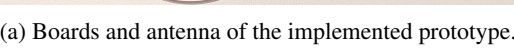

(a) Boards and antenna of the implemented prototype.

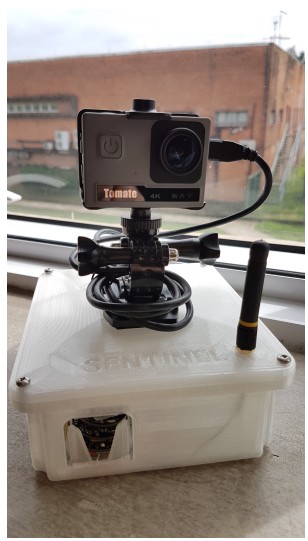

(b) The complete assembled prototype.

Figure 2: The implemented prototype used for the evaluation of the detection pipeline. All the pulga boards (base, core, LoRa and GPS) are assembled as a stack.

The Labrador, Pulga Stack, flat cable, and antenna are enclosed in a 3D-printed box with openings for interface connection and passive ventilation, as shown in 2. The USB Camera is mounted on top

---

[1]https://caninosloucos.org/en/labrador-64-en/

[2]Commonly called a "car cigarette lighter".

of the box, and fixed with a double-coated tape. The box is mounted on the middle of the top side of the vehicle panel, and the power cable is connected to the vehicular power outlet.

## 4.2 SOFTWARE

The detection Pipeline was implemented in Python and integrated within the `event_notifier`, a program responsible for managing the connection with the camera and other sensors, as well as starting and stopping the pipeline, and transmitting events and saving evidences.

Each step of the pipeline was implemented as a separate class, all of them implementing the method `run`, which receives the original frame and the result of the previous step. The pipeline was implemented as a separate class, which was in turn instantiated and called by a thread that manages the camera input, evidence storage, and the pipeline itself.

All steps use a YOLO object detection approach to perform inference, with the help of the `darknet` library. Each step uses a different `darknet` configuration composed of a network model file and a specific input image size. Varying the image size allows tuning the accuracy and execution time of the whole pipeline.

## 5 EVALUATION

The evaluation was conducted on the hardware specified in Section 4, namely the Labrador, a 1,3GHz quad-core single-board computer with 2 GB of volatile memory.

|  | Scan | Gun | Plate | Char |
|---|---|---|---|---|
| **Input image (pixels)** | 256 | 320 | 256 | 192 |
| **Execution time (s)** | 1.5 | 2.5 | 0.5 | 6 |

Table 1: The detection pipeline configuration and execution time.

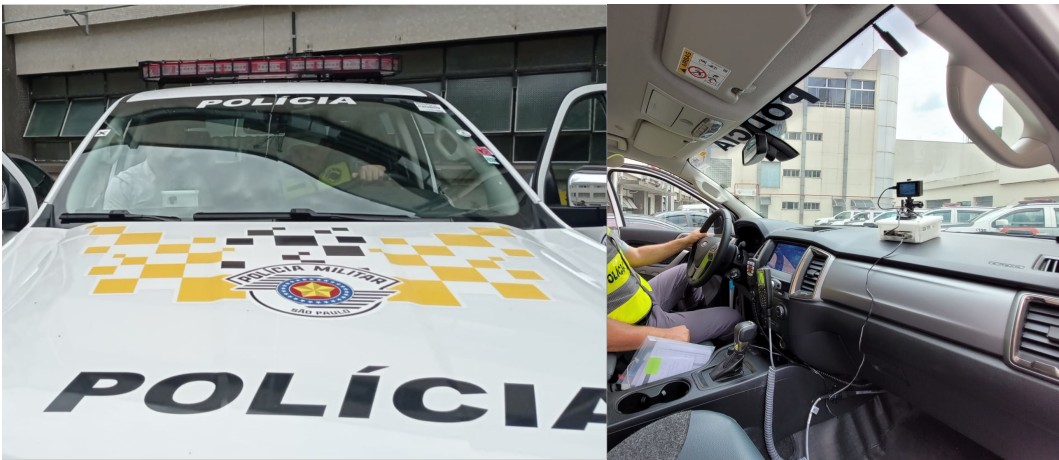

Figure 3: Deployment in Police patrol vehicles. The Sentinel device is powered by the car's battery and it is kept on whenever the vehicle's engine is also on.

In addition, the evaluation of the prototype devices employing our detection pipeline was conducted in partnership with the police department. Twelve devices were deployed in patrol vehicles for pilot testing rounds, where we mostly analysed license plate reading performance. From the preliminary results, we noticed that 78 % of the notifications received by the Police server were correct. With further analysis, we determined that this was due to a confusion between characters "5" and "6". After fixing this issue, detection accuracy increased to approximately 90 %. Examples of the geolocation data received are shown in 4a.

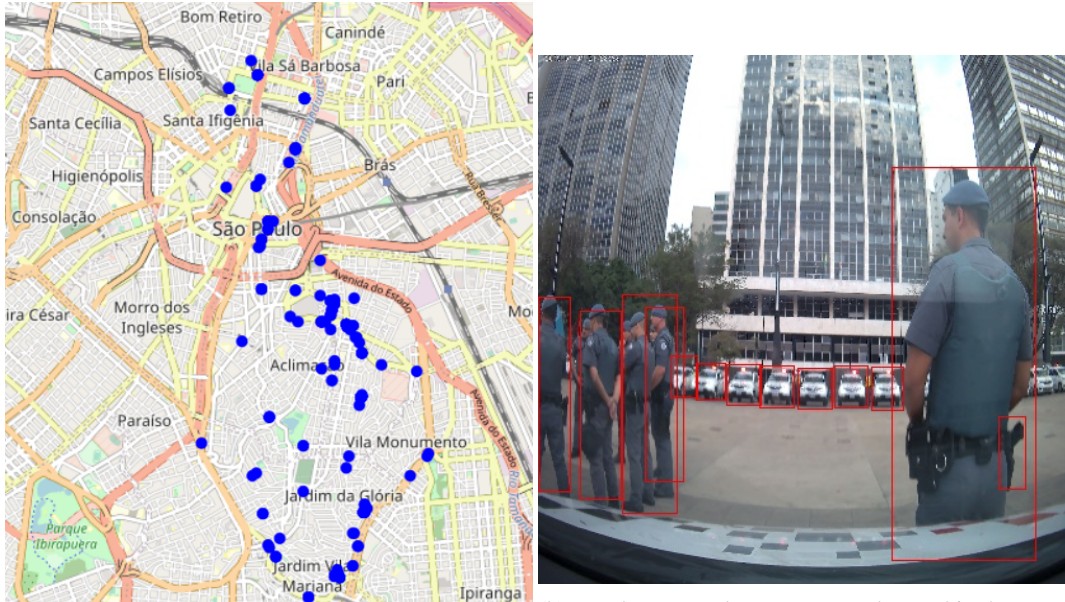

(a) Geolocation of some of the detected plates.

(b) People, cars and a weapon are detected in the same frame.

Figure 4: Geolocation of detected plates and an example of evidence frame stored in the prototype memory.

Image evidences of detected events are saved in the sentinel internal storage and can be latter retrieved manually. An example of detection frame is shown in 4b, where three categories of objects were correctly detected.

## 6   CONCLUSION

We introduced our pipeline-based approach for object detection in IoT devices. It was designed to leverage CV and AI methods with objects size and correlation to overcome the constrained resources of edge computing. We trained YOLO CNNs with our own built dataset and we evaluated our approach in a mobile surveillance system. Pilot testing was conducted in real world scenarios. The results shown that our approach is appropriate for IoT devices and edge computing, improving both the inference time and the mean average precision in object detection. Future works could include the evaluation of our approach in other object detection systems with even more constrained resources. In addition, it would be interesting to test the approach in a even larger quantity of smart IoT devices, as in a swarm paradigm.

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
