# OpenReview forum: "A Pipeline-Based Approach for Object Detection on Resource Constrained Internet of Things Devices"
_ICLR.cc/2024/Conference — Submitted to ICLR 2024_

### Official Review · Reviewer_cncP · 2023-10-30

**Soundness:** 1 poor
**Presentation:** 3 good
**Contribution:** 1 poor
**Rating:** 1
**Confidence:** 5

**Summary:**

The proposed work focuses on a smart surveillance application deployed on resource constraint devices with limited compute and memory capabilities. To address that, a pipeline-based object detection application is proposed. Specifically, size and correlation between detected objects is considered for passing to next phases of the pipeline. The goal for such approach is to improve the inference speed and reduce the number of false positives and negatives.

**Strengths:**

1) The problem of running advanced computer vision applications on resource constraints devices is very important especially in latency critical applications, where going to the cloud is not an option. Thus, the fact that this problem is emphasized in this work is really good.
2) Also, the method was tested in real application, operating in the field, showing its usefulness in the selected market vertical.
3) Description of the existing methods is very detailed with clear motivation for the work.

**Weaknesses:**

1) The proposed approach doesn't introduce anything novel to the ML research. Such implementation approach is pretty standard when deploying ML solutions in practice, and is tackling the software development portion of the application stack, not ML. Also, from the software implementation perspective the contribution is not very innovative, as other deployments usually use pipelined approached with model ensembles, loops and gating mechanisms when deciding what models to invoke.
2) The evaluation section is not comprehensive enough. The method should be compared with other techniques and not pipelined systems to justify the claims. Also, the abstract specifies that the inference time and mAP were improved, but it's not presented anywhere in the results section.

**Questions:**

It would be great if you could think about solutions that would optimize the ML model itself, instead of pipelining multiple models. One research direction could be creating edge-efficient ML architectures. Besides that, there are various techniques for making networks more suitable for edge devices, like quantization. I believe that contributing to that would be more impactful for the community like ICLR.

---

### Official Review · Reviewer_eUP2 · 2023-10-31

**Soundness:** 2 fair
**Presentation:** 2 fair
**Contribution:** 1 poor
**Rating:** 1
**Confidence:** 5

**Summary:**

The research paper introduces a pipeline-based approach for object detection on resource-constrained IoT devices, addressing the challenges associated with limited power and processing capacity. The proposed approach leverages computer vision and AI methods, considering object size and correlation to improve object detection in edge computing environments. The authors trained YOLO CNNs on their dataset and evaluated the approach in a mobile surveillance system. Pilot testing was conducted in collaboration with the police department, with a focus on license plate reading performance.

**Strengths:**

-	The paper addresses a significant issue concerning object detection on resource-constrained IoT devices in the context of smart surveillance systems. Ensuring safety in metropolitan areas is a pressing concern, and the proposed approach aligns with the needs of smart cities and law enforcement.
-	Collaborative testing with the police department in real-world scenarios adds credibility to the research and demonstrates the practical applicability of the proposed approach.

**Weaknesses:**

-	The work seems to be an implementation of already existing and popular YOLO architecture on custom dataset.
-	The work does not compare the work with other works that have been developed for surveillance.
-	The pipeline-based approach uses multiple runs of the YOLO model to detect objects which adds more time delay to detection missing other important potential events while processing one event.
-	While the paper provides some quantitative data, a more detailed analysis of the results, especially in the pilot testing, would enhance the paper's completeness.

**Questions:**

-	How many classes are present in the custom dataset being used in this work?
-	Is the model trained on edge device (Labrador)? Or was it just used for Inference?
-	What is the frame rate used in this methodology for detection?
-	What measures have been taken to detect occluded objects?
-	Since the use case is a mobile surveillance and involves testing in real world scenario, what are the processes in place to identify false events transmitted?
-	How do you keep up with the event generation when subsequent runs are needed? E.g. For generating an event from S3, you would need 8 seconds(1.5s+ 0.5s +6s). Does the system not detect any other events while it is processing one event?

---

### Official Review · Reviewer_SrVC · 2023-11-01

**Soundness:** 1 poor
**Presentation:** 2 fair
**Contribution:** 1 poor
**Rating:** 3
**Confidence:** 3

**Summary:**

Object detection on resource-constrained devices using computer vision and convolutional neural networks presents challenges due to their limited power and processing capabilities. The study introduces a multi-step detection pipeline tailored to the object sizes and their inter-correlations. When tested in a smart surveillance system incorporating edge computing and the IoT framework, and under real-world scenarios, this pipeline-based method tripled the processing speed and significantly enhanced the mean average precision of detection.

**Strengths:**

1. Followed by a very simple writeup, which is better for readability, specially for newer researchers.
2. The implemented hardware prototype the authors propose in Section 4.1.
3. This work has good real-life implementation.

**Weaknesses:**

1. The abstract is really small. It does not convey major information. For instance, “multi-step detection pipeline” - it does not have what this may do.
2. No research questions defined.
3. Too elaborated literature review on individual papers, but not enough number of papers in the lit review.
4. Too old models used for the AI pipeline. No proper explanation is also present why they choose older models instead of state-of-the-art models.
5. Even though the authors mention that they worked with four models, they did not mention which one they worked with mainly
6. No sample of data provided in paper.
7. Since the approach is worked on resource-constrained IoT devices, to compare, the authors did not bring any such metric (For instance, FLOPs, Parameters, Energy Consumptions) to compute the computational power needed by the models.
8. No result comparison is shown among the four models used.
9. Too less number of references used.
10. No limitation is present.
11. No discussion is present.

**Questions:**

1. Weakness #1: You should rework on the abstract. You should include more information, in a few words.
2. Weakness #2: In Section 1, you have added what are the challenges. However, please add what research questions you worked on here, in bullet points if possible.
3. To make your claims stronger, you should add some references on the statements being valid in Section 1.
4. You used very generalized words in section 1 that you will be using an AI based pipeline. You should add what the AI includes, and why the AI is needed to solve this, in a few words.
5. Section 2: Please shorten the similar type of research works under a subsection, in fewer words.
6. Section 2: Please add more research works.
7. I see a mixture of past and present tense in the sentences (For instance, in Section 1, 2). Also I saw using comma instead of full-stop in numbers (as decimals) (For instance, in Section 2). I would suggest you proofread the paper again.
8. Weakness #4: Please mention why you did not use YOLOv8 instead of 3 or 4 in the paper.
9. Weakness #5: You should add a few samples of the data.
10. “The goal for firearm detection was to achieve a mean average precision of (mAP) of at least 80 %.” - Is there any specific reason behind the number 80%? If yes, please mention it in paper.
11. It is not mentioned how much computational power is actually being taken by each YOLO model, in the device, and why you choose one specifically.
12. Please add the limitation subsection.
13. Please add the discussion section.  I would highly suggest adding it.

---

### Meta-Review · Area_Chair_32Cz · 2023-12-06

**Metareview:**

This paper introduces a multi-step object detection pipeline on resource-constrained devices with limited compute and memory capability.  While the paper addresses a significant issue regarding object detection on resource-constrained IoT devices and demonstrates the practical applicability of the proposed method in the real-world scenarios, the novelty is limited and the evaluation is not comprehensive enough.  The reviewers raised concerns on deployed old models of the pipeline without any justification.  The research question of this work was also unclear in the context of ML community.  Missing comparison with other works failed to convince the reviewers of the claims of the paper.  No rebuttal was submitted by the authors, unfortunately.  Therefore, the concerns raised by the reviewers have not resolved.  This paper should be rejected, accordingly.

**Justification For Why Not Higher Score:**

The reviewers unanimously agree to reject the paper.  No rebuttal was submitted.

**Justification For Why Not Lower Score:**

N/A

---

### Decision · Program_Chairs · 2024-01-16

Reject